# Zero-shot Generalizable Incremental Learning for Vision-Language Object Detection

Jieren Deng[1,2], Haojian Zhang [*1], Kun Ding[1], Jianhua Hu[1], Xingxuan Zhang[3], and Yunkuan Wang[1]

[1]Institute of Automation, Chinese Academy of Sciences (CAS)
{dengjieren2019, jianhua.hu, zhanghaojian2014, yunkuan.wang}@ia.ac.cn
[2]School of Artificial Intelligence, University of Chinese Academy of Science, UCAS
[3]Shanghai Sixth People's Hospital Affiliated to Shanghai Jiao Tong University School of Medicine
zhangxingxuan@sjtu.edu.cn

## Abstract

This paper presents Incremental Vision-Language Object Detection (IVLOD), a novel learning task designed to incrementally adapt pre-trained Vision-Language Object Detection Models (VLODMs) to various specialized domains, while simultaneously preserving their zero-shot generalization capabilities for the generalized domain. To address this new challenge, we present the Zero-interference Reparameterizable Adaptation (ZiRa), a novel method that introduces Zero-interference Loss and reparameterization techniques to tackle IVLOD without incurring a significant increase in memory usage. Comprehensive experiments on COCO and ODinW-13 datasets demonstrate that ZiRa effectively safeguards the zero-shot generalization ability of VLODMs while continuously adapting to new tasks. Specifically, after training on ODinW-13 datasets, ZiRa exhibits superior performance compared to CL-DETR and iDETR, boosting zero-shot generalizability by substantial **13.91** and **8.74** AP, respectively. Our code is available at https://github.com/JarintotionDin/ZiRaGroundingDINO.

## 1 Introduction

Object detection has achieved remarkable strides in recent years [2, 44]. However, most object detection models are typically trained to predict a predefined, closed set of categories [32, 2], constraining their prospect in real-world applications. Recent research endeavors [12, 27, 45, 19, 22] focus on developing intelligent systems capable of detecting objects specified by natural language inputs, giving rise to the field of Vision-Language Object Detection (VLOD) (also named Open Vocabulary Object Detection), also referred to as Open Vocabulary Object Detection. In this framework, the models are termed Vision-Language Object Detection Models (VLODMs). By incorporating natural language inputs, VLODMs can recognize a much broader set of visual concepts beyond a fixed category set, resulting in exceptional zero-shot generalizability.

Despite VLODMs' great zero-shot recognition ability in the general domain, VLODMs often exhibit suboptimal performance in more specialized domains, such as identifying aquatic organisms in aquariums or interpreting remote sensing images from aerial drones. In real-world scenarios, the necessity to adapt VLODMs to various unforeseen downstream tasks remains crucial in order to attain the desired accuracy, as highlighted in recent research [19, 45].

A straightforward idea is to adapt different individual VLODMs to different downstream tasks. However, a general agent often needs to simultaneously recognize objects from diverse tasks, each

---

*Corresponding author

38th Conference on Neural Information Processing Systems (NeurIPS 2024).

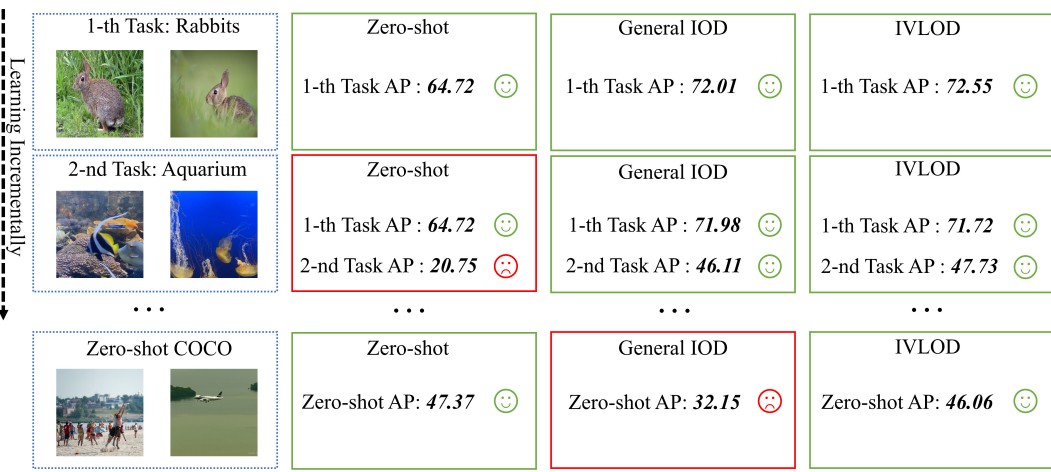

Figure 1: Incremental Vision-Language Object Detection (IVLOD) aims to enhance VLODMs' performance across specialized domains via incremental learning, while also preserving their zero-shot generalization capability, enabling them to handle both known and unknown objects simultaneously and effectively.

learned at distinct times. Moreover, in a dynamically changing environment, equipping VLODMs with the ability of incremental learning is paramount [20, 31], considering that downstream tasks typically arrive in a streaming manner and incremental learning paradigm better aligns with the learning and cognition processes of a general intelligent agent.

To endow VLODMs with the incremental learning ability, a naive idea is applying existing methods [10, 23] designed for detector on closed set directly. However, from our observations, it will weaken the excellent zero-shot generalization ability of VLODMs to detect unseen categories. To highlight the distinctiveness of incremental learning for VLODMs, we believe it is necessary to introduce a new task, Incremental Vision-Language Object Detection (IVLOD), which stresses the importance of maintaining the zero-shot generalization ability of VLODMs while adapting to various downstream tasks incrementally.

To further demonstrate the uniqueness of the IVLOD task, we have compared three different variants of adapting VLODMs to multiple downstream tasks: zero-shot learning (*i.e.*, no adaptation), conventional incremental learning (applying CL-DETR [23] to VLODM), and zero-shot generalizable incremental learning (*i.e.*, the IVLOD task) in Fig. 1. Unlike general incremental object detection (general IOD, ref. second column), IVLOD is characterized by preserving the original zero-shot performance of VLODM while also performing incremental learning. From this comparison, IVLOD faces two major challenges. One is the issue known as catastrophic forgetting [29], where the performance on previously learned tasks may sharply decline when new tasks are introduced. The other challenge is to maintain the zero-shot generalizability of VLODMs while learning new downstream tasks.

From a unified perspective, both challenges can be attributed to the forgetting problem. Technically, two kinds of methods can tackle this issue: selecting exemplar samples from past datasets for replaying [31, 23] or maintaining a duplicate model to facilitate knowledge distillation [20, 23]. In order to prevent forgetting, the replaying-based approach necessitates preserving enough examples to ensure the representativeness of the samples for replaying, but it is hard to do this with a limited-size memory, specifically when the model is pre-trained on a large-scale vision-language dataset. At the same time, current knowledge distillation techniques do not adequately focus on preventing the forgetting of pre-training knowledge, rendering them suboptimal for IVLOD tasks. Furthermore, they often need to store an entire model copy, which requires a large memory budget. Such storage demands are often untenable, particularly in scenarios that use resource-constrained edge devices.

To effectively address IVLOD's challenges while improving memory efficiency, we introduce a novel approach named Zero-interference Reparameterizable Adaptation (ZiRa). ZiRa retains the parameters of the original model and introduces a parallel dual branch structure named Reparameterizable Dual Branch (RDB) designed for efficient tuning on downstream tasks. The RDB structure is

reparameterizable, allowing the model to adapt to downstream tasks without increasing memory usage. Furthermore, the RDB structure serves a dual purpose—it achieves the branch labor division for downstream continual learning, protecting the learned knowledge from being excessively overwritten, and more crucially, it lays the structural foundation for the key novel element of ZiRa named Zero-interference Loss (ZiL).

ZiL simultaneously penalizes the output norms of both the entire RDB and a high-learning-rate branch within the RDB, guiding the RDB to learn in a direction that protects both knowledge learned from pre-training and downstream tasks. The idea of ZiL comes from the following insight: During pre-training, the model is trained with visual-language inputs that contain significant noise, and the model has learned how to handle this noise. As a result, the pre-trained VLODMs have a certain degree of robustness to the input, and they can handle a certain range of interference without their performance being affected. ZiL ensures that the fine-tuned residual input of RDB is of small norm, thereby guaranteeing that the model's original performance. At the same time, for adaptation, only a small adjustment to the input is needed to learn new concepts. Although ZiL constrains the input norm of the RDB, it still allows the RDB to acquire sufficient downstream knowledge. As a result, ZiL effectively addresses the two challenges of IVLOD at the same time. Crucially, ZiRa accomplishes this without necessitating the duplication of the entire model or any exemplars, thereby ensuring efficient utilization of memory.

Compared to existing methods [34, 23], ZiRa leverages the advantages offered by VLODMs more fully through a principled design (*i.e.*, the ZiL based on RDB), offering a superior and more effective solution without the need for excessive additional memory. The contributions of this paper can be summarized as follows:

- We introduce Incremental Vision-Language Object Detection (IVLOD), a novel learning task that incrementally adapts VLODMs to multiple specialized domains while preserving zero-shot generalization ability.

- We present a novel approach called ZiRa to tackle IVLOD challenges. Based on the reparameterizable parallel tuning structure of RDB, ZiRa introduces ZiL to simultaneously minimize interference with zero-shot performance and prevent forgetting on downstream tasks. Notably, ZiRa makes these achievements in a memory-efficient manner.

- We conduct comprehensive experiments on COCO and ODinW-13 datasets, showcasing the effectiveness of our method. For instance, ZiRa outperforms CL-DETR [23] and iDETR [10] by a significant margin, making a **13.91** and **8.74** improvement in zero-shot AP on COCO, respectively, while offering competitive results on downstream tasks incremental learning.

## 2  Related Work

**Vision-Language (or Open Vocabulary) Object Detection.** Vision-Language (or Open Vocabulary) Object Detection Models combine natural language descriptions with object detection in images. Early research in this area primarily focused on grounding textual phrases to image regions, such as grounding referring expressions [28, 39, 4] and visual question answering [1, 25, 42]. More recent studies have developed unified and scalable approaches for simultaneous object detection and grounding tasks. For instance, GLIP [19] unifies phrase grounding and object detection tasks by treating object detection as context-free phrase grounding and treating phrase grounding as a contextualized object detection task. This approach enables the integration of multi-layered textual information into the detection model. Grounding DINO [22], based on the DINO [44], employs a DETR-like [2] architecture that uses a Transformer as the detector, which allows for unified processing of image and language data and offers the outstanding capacity to utilize large-scale datasets. Based on Grounding DINO, our work further addresses the newly introduced IVLOD tasks.

**Incremental Learning and Incremental Object Detection.** Incremental learning, also known as continual learning or lifelong learning, tackles the challenge of learning new tasks without forgetting previous ones. Incremental learning has given rise to two primary strategies, memory replay, and knowledge distillation, to tackle its inherent challenges [20, 31, 3, 5, 38, 37, 40]. For example, Learning without Forgetting (LwF) [20] employs knowledge distillation to maintain performance on old tasks while learning new ones. Another notable approach, iCaRL [31], combines exemplar memory

and a nearest-mean-of-exemplars classification method for incremental learning. Incremental object detection, a specialized subfield within incremental learning, focuses on improving the adaptability of object detection models to new object categories while maintaining performance on previously learned categories [34, 17, 30, 41, 11, 10, 23]. ILOD [34] was the first to introduce incremental object detection. More recently, CL-DETR [23] effectively combined knowledge distillation and exemplar memory techniques to enhance incremental object detection using Transformer-based models. Our work distinguishes itself by performing incremental learning on VLODMs, which are more favorable for open-world problems. Besides the need to continually adapt across multiple specialized tasks, it is also important to preserve their zero-shot generalization ability. To the best of our knowledge, our work is the first to pose and tackle IVLOD tasks.

**Open-World Object Detection.** Open-World Object Detection (OWOD) [16, 13, 26] also targets the detection of both seen and unseen objects, but it involves simultaneous learning of new objects and detecting unknown objects during the incremental learning process. This means that as new tasks are introduced, the model is trained to recognize new objects while also developing its ability to detect unknown objects. In contrast, Incremental Vision-Language Object Detection (IVLOD) pre-trains models to detect unknown objects before the incremental learning process begins. The focus of IVLOD during incremental learning is on preserving the model's ability to detect these unknown objects while adapting to new tasks. Given the superior zero-shot generalization capabilities of current vision-language models, we believe that starting with a pre-trained vision-language model for IVLOD is a more effective approach compared to the traditional OWOD paradigm.

**Reparameterization.** Reparameterization techniques optimize model inference speed by transforming the training-time architecture into an equivalent, simplified form for inference [8, 9, 7]. RepVGG [9] exemplified this approach by employing $3 \times 3$ and $1 \times 1$ convolutional branches for training, which are streamlined to $3 \times 3$ convolutional branches and ReLU activations during inference. RepMLPNet [7] introduced "Locality Injection" to merge trained parameters of parallel convolutional kernels into fully connected layers, enhancing MLP-based model performance. Our work builds upon the Reparameterizable Dual Branch (RDB) concept initially proposed by [43]. The key innovations introduced in our approach include the use of differentiated learning rates for the High Learning Rate Branch and Low Learning Rate Branch, and the implementation of Zero-interference Loss (ZiL). These enhancements allow for a more effective balance between learning new tasks and preserving previously acquired knowledge, distinguishing our method from prior work.

## 3 Methodology

In this section, we systematically present our approach. We begin with an overview (Sec. 3.1), and then introduce Zero-interference Reparameterizable Adaptation (ZiRa), which includes the Reparameterizable Dual Branch (RDB) (Sec. 3.2) and the Zero-interference Loss (ZiL) (Sec. 3.3).

### 3.1 Overview

The framework of our model for addressing the IVLOD task is illustrated in Fig. 2, with the selected VLODM being Grounding DINO [22]. Other VLODMs with a similar structure are also suitable. Specifically, the VLODM takes image features and category text prompts as inputs and produces class-agnostic proposals whose features are closely aligned with the text prompts. These text prompts are also refined to obtain text classification features by fusing the image features. Subsequently, the model classifies the class-agnostic proposals based on the cosine similarity between the visual features of proposals and the text classification features. These VLODMs are pre-trained on extensive grounding datasets annotated with detailed human language and object boxes, which equips them with great zero-shot recognition capabilities. To address the challenges of IVLOD, we propose ZiRa which is based on the structure of RDB and constrains the output of RDB by ZiL.

### 3.2 Reparameterizable Dual Branch

**Adaption on Both Language and Vision Sides.** Fine-tuning the entire model is a common method for adapting the original model to downstream tasks. However, it has been demonstrated that tuning only a few newly introduced parameters in a pre-trained model is more effective [14, 15], while full fine-tuning often leads to significant forgetting. In view of this, we introduce additional parallel

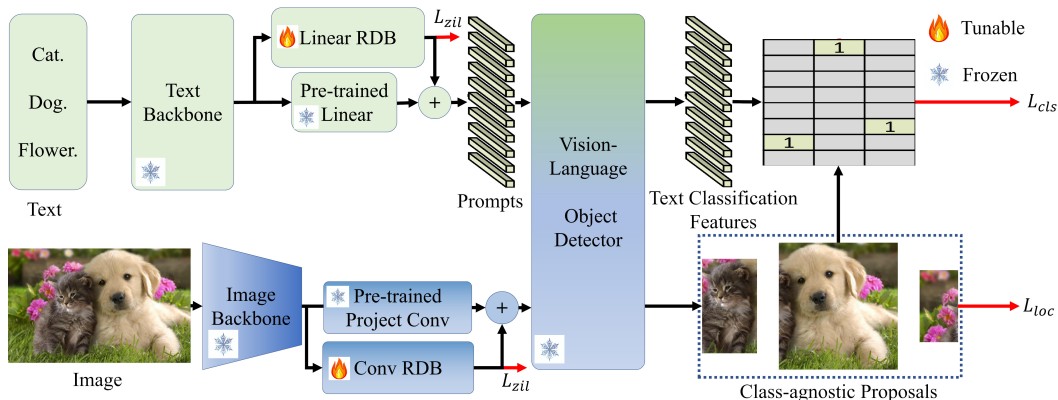

Figure 2: Our framework, features two Reparameterizable Dual Branch with Zero-interference Loss on both the vision and language sides.

branches on both the language and vision sides of the VLODM to adapt the model for sequential downstream tasks.

As shown in Fig. 2, these additional branches are termed the Reparameterizable Dual Branch (RDB). On the language side, we integrate an RDB alongside a pre-trained linear layer positioned between the detector and the language backbone network, aiming to learn the new high-level semantic concepts. On the visual side, we also connect a supplementary RDB in parallel with a pre-trained convolutional layer located between the visual backbone and the detector, aiming to learn the visual features of new classes. It's pivotal to adapt on both sides, as these two sides capture distinct structures of vital knowledge and the lack of either will lead to insufficient adaptation to downstream tasks.

**Dual Branch Structure within the RDB.** Within the RDB, illustrated in Fig. 3, our design features a dual-branch structure, comprising the Low-learning rate Branch (LLRB) and the High-learning rate Branch (HLRB). The HLRB is set to a high learning rate for rapid task adaptation, whereas the LLRB employs a more conservative learning rate. Specifically, the LLRB is set at $\eta$ $(0 < \eta < 1)$ times the learning rate of the HLRB. This differentiation in learning rates brings division of labor between LLRB and HLRB, especially when combined with our next strategy that reparameterizes the HLRB into the LLRB after learning each task. The LLRB's low learning rate can help maintain downstream task knowledge, while the HLRB's high learning rate can help swift adaptation to new tasks. This division protects the knowledge stored in the LLRB from being excessively overwritten when incrementally learning downstream tasks, achieving a better balance between stability the plasticity, while the single branch or naive dual branch structure can not take advantage of this division mechanism.

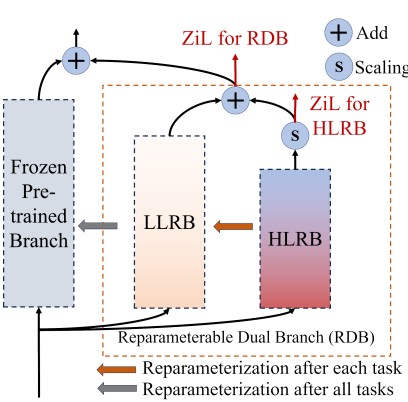

Figure 3: The structure of the Reparameterizable Dual Branch (RDB).

Specifically, the collective output of the RDB $x_{rdb}$, is formulated as:

$$x_{rdb} = \text{HLRB}(x) \cdot s + \text{LLRB}(x), \qquad (1)$$

where $x$ is the input and $s$ is a learnable scaling factor.

For the language side, the RDB is structured as two parallel linear layers, and its output $x^l_{rdb}$ is expressed as:

$$x^l_{rdb} = (W^l_{hlrb}x + b^l_{hlrb}) \cdot s + W^l_{llrb}x + b^l_{llrb}, \qquad (2)$$

where $W^l_{hlrb}$ and $W^l_{llrb}$ represent the weights of HLRB and LLRB, respectively, while $b^l_{hlrb}$ and $b^l_{llrb}$ denote their corresponding biases.

For the vision side, the RDB is structured as two parallel convolution layers, and its output $x_{rdb}^v$ is illustrated as:

$$x_{rdb}^v = (W_{hlrb}^v \otimes x + b_{hlrb}^v) \cdot s + W_{llrb}^v \otimes x + b_{llrb}^v. \tag{3}$$

Here, $W_{hlrb}^v$ and $W_{llrb}^v$ stand for the kernel of HLRB and LLRB, respectively, while $b_{hlrb}^v$ and $b_{llrb}^v$ represent their corresponding biases. $\otimes$ denotes the convolution operation.

**Reparameterization from the HLRB to the LLRB.** With each new task, ZiRa resets the HLRB parameters to zero and, after completing learning on a new task, merges the trained parameters from the HLRB into the LLRB. This approach not only allows the LLRB to incrementally acquire new knowledge but also avoids the linear increase in the number of additional branches as the number of tasks increases, effectively managing memory consumption. Moreover, it offers a structural prerequisite for ZiL to prevent forgetting on downstream tasks by penalizing the output norm of HLRB, which will be detailed in the next section.

On the language side, ZiRa combines the HLRB and LLRB into a consolidated LLRB with new weights $W_{hlrb}^l \cdot s + W_{llrb}^l$ and new biases $b_{hlrb}^l \cdot s + b_{llrb}^l$, given by:

$$x_{rdb}^l = (W_{hlrb}^l \cdot s + W_{llrb}^l)x + (b_{hlrb}^l \cdot s + b_{llrb}^l). \tag{4}$$

Similarly, on the vision side, the HLRB and LLRB are merged into a singular LLRB, as depicted by:

$$x_{rdb}^v = (W_{hlrb}^v \cdot s + W_{llrb}^v) \otimes x + (b_{hlrb}^v \cdot s + b_{llrb}^v), \tag{5}$$

where $W_{hlrb}^v \cdot s + W_{llrb}^v$ and $b_{hlrb}^v \cdot s + b_{llrb}^v$ can be considered as the new kernel and new biases.

**Single Branch for Inference.** Sub-branches inside the RDB like the LLRB and HLRB can increase computational overhead. We efficiently address this by reparameterizing all sub-branches in RDB and the frozen pre-trained branch into a new one for inference. Since the frozen pre-trained branch is also a linear structure, after learning all the tasks, we can create a new branch that merges the frozen pre-trained branch and the RDB via reparameterization. During inference, we exclusively use this newly consolidated branch, but we still retain the parameters frozen pre-trained branch and the RDB for future learning. This approach significantly reduces the computational cost during inference, as only one branch is required for prediction

### 3.3 Zero-interference Loss

**Protect Zero-shot Generalizability with ZiL.** Preventing performance degradation on the original pre-training domain can be viewed as preventing the VLODM from forgetting the original pre-training domain knowledge. Two common methods to avoid forgetting is to reserve some old samples for replaying or distill the current model from the previous ones. However, preserving old samples and storing the previously trained model requires significant additional memory. On the contrary, Zero-interference Loss (ZiL) can prevent forgetting in a memory-efficient manner.

Specifically, ZiL penalizes the norm of the whole RDB's output (ref. Fig. 3), aiming at safeguarding the zero-shot performance of VLODM. The ZiL for the RDB, $L_{rdb}$, is defined as follows:

$$L_{rdb} = L_1(x_{rdb}). \tag{6}$$

Here, $L_1$ corresponds to the $L_1$ norm.

**Why ZiL works.** The effectiveness of ZiL can be explained as follows. First, the VLODM's input (What we fine-tuned in this paper) is robust to small-norm random perturbations, which have little impact on the model's performance. To verify this, we add Gaussian noise to the input of the pre-trained VLODM's detector and observe the model's performance. As shown in Fig. 4, the model's performance is not significantly affected by the addition of small-norm random noise to the input.

Second, the ZiL ensures that the fine-tuned input of the VLODM has a small norm additional term. To further investigate the effectiveness of ZiL, we plot curves illustrating the evolution of the RDB's output norm throughout the IVLOD process, as shown in Fig. 5. It is evident that training the RDB with ZiL leads to a substantial reduction in the output norm of the RDB on COCO, compared to training the RDB without ZiL. This preliminary observation strongly suggests that ZiL effectively preserves the zero-shot generalizability of VLODMs. Examining the evolution curves in Fig. 5 more closely, we observe that in the initial stages, ZiL plays a relatively minor role in curbing the growth

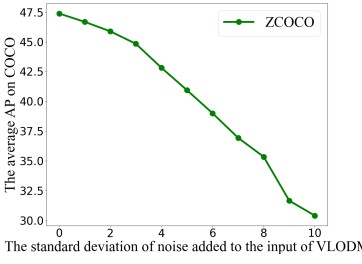

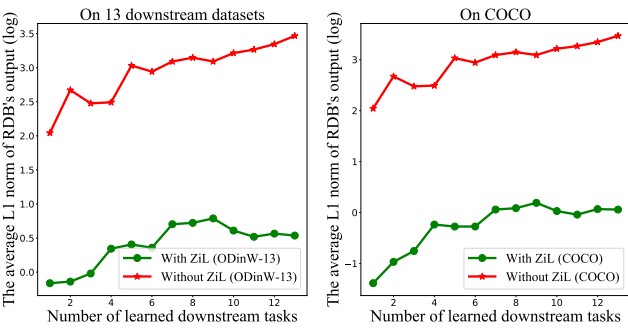

Figure 4: The performance of the pre-trained VLODM with different levels of Gaussian noise added to the input of VLODM's detector.

Figure 5: The average $L_1$ norm curve of the RDB's output overall sequentially learned downstream tasks, computing the output norm on both language and vision sides. The longitudinal axis is logarithmically scaled for better visualization.

of the RDB's output norm. However, as the process of accumulating new knowledge progresses, the influence of ZiL becomes more significant, leading to stronger constraint effects. This increase in ZiL's impact leads to a dynamic balance: the interference caused by integrating new knowledge is counteracted by the interference reduction achieved by ZiL, which is reflected in the fact that as the number of tasks increases, the norm of the RDB's output doesn't rise unboundedly but rather stabilizes after learning a certain number of tasks. This dynamic balance is crucial for maintaining the zero-shot generalizability of VLODMs.

**Prevent forgetting on downstream tasks with ZiL.** The $L_{rdb}$ serves primarily to preserve the zero-shot generalization capability, leaving the challenge of downstream task forgetting unresolved. By exploiting the advantage of RDB, we address this challenge by further applying ZiL on HLRB, which is computed as:

$$L_{hlrb} = L_1(\text{HLRB}(x) \cdot s), \tag{7}$$

where $s$ is a learnable scaling parameter.

The $L_{hlrb}$ can effectively protect the knowledge that is learned from previous tasks and encompassed within the combined LLRB and the pre-trained branch. Specifically, given that the knowledge learned from the previous task has already been reparameterized into LLRB and LLRB learns at a lower learning rate, the total output of the branches previously trained can be regarded as $\text{LLRB}(x) + \text{PTB}(x)$, where PTB is the pre-trained branch. The total output of the current branches is $\text{LLRB}(x) + \text{HLRB}(x) \cdot s + \text{PTB}(x)$, which has an additional component $\text{HLRB}(x) \cdot s$ compared to the output from the branches trained on the previous task. Hence, optimizing $L_{hlrb}$ equates to penalizing the discrepancy in the outputs between the models trained on the previous and the current task, effectively guiding the HLRB to learn new tasks without much downstream task forgetting.

**Total Optimization Objective.** For a general VLODM, we continue to employ its original training loss. To prevent forgetting, we incorporate an additional loss term, $L_{zil}$. Taking Grounding DINO as an example, the total loss function becomes:

$$L_{total} = L_{cls} + L_{loc} + \lambda L_{zil}, \tag{8}$$

where $L_{zil} = L_{rdb} + L_{hlrb}$ is the proposed ZiL, $\lambda$ is its loss weight, and $L_{cls}$ and $L_{loc}$ are the contrastive classification loss and localization loss of Grounding DINO. Throughout the training process, the entire original VLODM is frozen, except for the RDB.

## 4 Experiments

### 4.1 Setup

**Datasets.** We conduct our experiments on the COCO [21] datasets and the "Object Detection in the Wild (ODinW)" [18] benchmark. ODinW is a more challenging benchmark designed to test model performance under real-world scenarios. It comprises numerous sub-datasets from various domains for evaluation, such as Thermal (to detect objects in heat map images) and Aquarium (to detect different marine animals). Following GLIP [19], we use ODinW-13 datasets, they are labeled as Ae

(Aerial Maritime Drone), Aq (Aquarium), Co (Cottontail Rabbits), Eg (Egohands), Mu (Mushrooms), Pa (Packages), Pv (Pascal VOC), Pi (Pistols), Po (Pothole), Ra (Raccoon), Sh (Shellfish), Th (Thermal Dogs and People), Ve (Vehicles). The 13 sub-datasets of ODinW-13 are trained sequentially, one by one, and are tested after all sub-datasets have been trained.

**Metrics.** As COCO is a dataset within the realm of everyday life, we use zero-shot COCO performance as a metric of the model's zero-shot generalization ability across general domains. Additionally, we provide the average performance on the downstream ODinW-13 benchmark, with performance calculations made only after the model has been sequentially trained on all sub-datasets. We denote the zero-shot mAP on COCO as **"ZCOCO"** and the average mAP on ODinW-13 downstream tasks as **"Avg"**.

**Implementation Details.** In this paper, unless otherwise specified, we use the **Grounding DINO** [22] with the Swin-Tiny [24] backbone as the default original VLODM in our experiments. This Grounding DINO is pre-trained on Objects365 [33], GoldG [19], and Cap4M [19]. We also use OV-DINO [35] with the Swin-Tiny backbone as an additional validation to demonstrate ZiRa's ability to generalize effectively across different DETR-based architectures. This OV-DINO model is pre-trained on the Objects365 [33], GoldG [19], and Cap1M [35] datasets. The learning order of the ODinW-13 sub-datasets is randomly shuffled. All results are obtained after running experiments three times with different random seeds. The hyperparameter experiments are also conducted on the ODinW-13 dataset and COCO, and the results are shown in the appendix.

Our proposed method is implemented with PyTorch and trained on two Nvidia RTX 3090 GPUs. Each downstream task is trained for a total of two epochs with a batch size of 2. For Grounding DINO, we employ an initial learning rate of $10^{-3}$, which decays to 0.1 times the original value after the first epoch to ensure effective convergence. For OV-DINO, we employ an initial learning rate of $10^{-4}$, which also decays to 0.1 times the original value after the first epoch to ensure effective convergence. AdamW is used as the optimizer, and the weight decay is $10^{-4}$. At the beginning of IVLOD, we initialize the parameters of the total RDB as zero.

Table 1: IVLOD results on ODinW-13 and ZCOCO. All results are based on the same Grounding DINO. "Shots" means how many samples are used when adaptation training.

| Shots | Methods | ZCOCO | Avg | Ae | Aq | Co | Eg | Mu | Pa | Pv | Pi | Po | Ra | Sh | Th | Ve |
|---|---|---|---|---|---|---|---|---|---|---|---|---|---|---|---|---|
| 0 | Original Model | 47.37 | 46.70 | 19.10 | 20.75 | 64.72 | 56.96 | 25.39 | 54.49 | 54.80 | 65.97 | 22.08 | 62.22 | 32.84 | 70.62 | 57.15 |
| 1 | TFA | 18.84 | 39.50 | 18.25 | 15.81 | 63.90 | 50.79 | 28.47 | 50.37 | 29.49 | 59.16 | 21.90 | 50.67 | 19.86 | 60.85 | 43.97 |
| | iDETR | 44.61 | 49.82 | 22.81 | 23.24 | 69.75 | 61.43 | 31.73 | 56.27 | 55.40 | 62.44 | 28.45 | 60.33 | 43.33 | 73.64 | 58.84 |
| | AT | 44.11 | 46.23 | 21.55 | 23.62 | 66.60 | 58.96 | 27.68 | 53.97 | 54.58 | 62.47 | 26.94 | 53.17 | 20.37 | 70.31 | 60.71 |
| | ZiRa | 45.25 | 50.20 | 19.91 | 24.84 | 68.82 | 63.02 | 37.02 | 60.20 | 55.11 | 63.37 | 28.26 | 66.35 | 38.97 | 69.27 | 57.43 |
| 5 | TFA | 26.43 | 45.76 | 21.92 | 22.30 | 67.40 | 60.72 | 30.63 | 53.56 | 46.80 | 63.60 | 26.88 | 56.26 | 28.00 | 64.28 | 52.49 |
| | iDETR | 43.51 | 51.65 | 25.69 | 25.53 | 70.42 | 62.98 | 49.98 | 50.54 | 54.85 | 64.80 | 33.24 | 57.64 | 42.36 | 76.51 | 56.92 |
| | AT | 43.67 | 47.16 | 14.63 | 24.97 | 66.56 | 64.19 | 38.85 | 42.03 | 55.49 | 65.20 | 27.48 | 52.68 | 32.21 | 71.23 | 57.54 |
| | ZiRa | 45.53 | 54.19 | 24.13 | 30.92 | 72.46 | 66.34 | 51.23 | 56.27 | 60.35 | 67.44 | 36.22 | 60.18 | 42.22 | 77.18 | 59.55 |
| 10 | TFA | 34.35 | 46.61 | 21.17 | 22.16 | 66.82 | 60.63 | 32.35 | 50.15 | 55.54 | 64.98 | 27.59 | 57.40 | 28.14 | 66.82 | 52.11 |
| | iDETR | 43.54 | 53.29 | 25.39 | 27.70 | 65.62 | 67.58 | 47.99 | 60.20 | 56.32 | 63.93 | 35.18 | 59.37 | 53.63 | 74.70 | 55.12 |
| | AT | 43.06 | 47.34 | 18.73 | 25.42 | 69.77 | 66.34 | 35.84 | 48.25 | 53.39 | 64.07 | 28.89 | 50.33 | 32.50 | 66.57 | 55.37 |
| | ZiRa | 45.70 | 54.86 | 25.01 | 29.44 | 69.59 | 68.44 | 54.72 | 61.68 | 59.69 | 67.14 | 36.05 | 61.77 | 48.30 | 73.92 | 57.47 |
| Full | TFA | 30.97 | 47.93 | 23.80 | 30.65 | 67.21 | 61.77 | 30.52 | 50.23 | 47.73 | 60.91 | 29.25 | 61.72 | 31.42 | 66.23 | 61.61 |
| | iDETR | 37.32 | 58.71 | 32.64 | 46.65 | 70.99 | 68.56 | 55.32 | **58.88** | 64.48 | 71.01 | **50.33** | 63.30 | 39.19 | **77.12** | 64.80 |
| | AT | 42.30 | 51.14 | 23.62 | 39.90 | 72.32 | 65.51 | 31.47 | 50.48 | 60.51 | 66.07 | 39.09 | 53.50 | 34.04 | 68.07 | 60.23 |
| | OW-DETR | 31.22 | 55.58 | 28.46 | 43.78 | 70.54 | 67.78 | 43.84 | 56.75 | 63.13 | 69.51 | 45.16 | 58.99 | 36.99 | 74.42 | 63.20 |
| | CL-DETR | 32.15 | 57.26 | 29.35 | 45.15 | **71.94** | **69.90** | 45.21 | 58.52 | **65.10** | **71.68** | 46.58 | 60.83 | 38.14 | 76.74 | **65.18** |
| | ZiRa | **46.06** | **59.73** | **32.81** | **48.19** | 70.33 | 69.67 | **59.33** | 58.05 | 64.04 | 70.67 | 50.06 | **67.49** | **45.51** | 76.76 | 63.54 |

Table 2: IVLOD results on ODinW-13 and ZCOCO. All results are based on the same OV-DINO.

| Shots | Methods | ZCOCO | Avg | Ae | Aq | Co | Eg | Mu | Pa | Pv | Pi | Po | Ra | Sh | Th | Ve |
|---|---|---|---|---|---|---|---|---|---|---|---|---|---|---|---|---|
| Zero | Original Model | 50.22 | 26.64 | 15.69 | 19.37 | 11.79 | 40.67 | 1.23 | 59.26 | 50.78 | 12.19 | 2.44 | 33.81 | 8.50 | 43.68 | 46.86 |
| Full | TFA | 44.98 | 47.56 | 22.27 | 35.36 | 63.58 | 56.68 | 19.84 | 64.26 | **68.07** | **50.01** | 25.99 | 56.31 | 33.70 | 56.96 | **65.20** |
| Full | CL-DETR | 34.52 | 45.28 | 23.07 | 27.90 | 46.81 | 67.72 | 25.39 | 66.56 | 59.12 | 28.04 | 29.48 | 64.71 | 25.58 | 68.28 | 56.05 |
| Full | AT | 36.80 | 44.33 | 22.46 | 27.42 | 45.78 | 67.10 | 24.10 | 65.08 | 58.17 | 26.65 | 28.88 | 64.46 | 24.60 | 67.30 | 54.27 |
| Full | iDETR | 37.71 | 46.91 | **24.09** | 30.56 | 49.16 | **68.49** | **26.05** | 67.45 | 61.85 | 30.75 | 29.50 | **66.83** | 27.11 | **71.27** | 56.69 |
| Full | ZiRa | **49.07** | **50.21** | 23.37 | **40.43** | **69.02** | 66.21 | 20.74 | 58.51 | 67.74 | 46.71 | **34.13** | 64.62 | **39.44** | 58.01 | 63.81 |

## 4.2 Comparison with Existing Methods

In our evaluation, we compare ZiRa with existing Incremental Object Detection (IOD) methods, with a specific focus on methods based on Detection Transformers (DETR) [2], as our approach falls into this category. The baseline methods we consider include:

- TFA [36]: A classical linear-probing-based incremental few-shot object detection baseline.
- OW-DETR [13]: An open-world object detection approach based on DETR, primarily addressing forgetting through exemplar replaying, which necessitates additional memory to store exemplars.
- CL-DETR [23]: A DETR-based method employing refined knowledge distillation and exemplar replaying. This method requires a complete model copy and exemplars, incurring significant extra memory costs.
- iDETR [10]: An incremental few-shot object detection method specifically focusing on tuning the projection layer of DETR-like detectors via knowledge distillation and self-supervision.
- Adapting-tuning (AT) [14]: A parameter-efficient adaptation method that has demonstrated its effectiveness in incremental few-shot object detection [6].

We first compare the above methods implemented on the Grounding DINO model. Experiments are conducted under both few-shot and full-shot settings. Since CL-DETR and OW-DETR are not designed for few-shot IOD, we only compare them under the full-shot setting. The results presented in Tab. 1 demonstrate that ZiRa consistently outperforms existing IOD methods in terms of "Avg" on downstream tasks. Moreover, ZiRa exhibits remarkable performance in preserving the zero-shot generalization ability of VLODMs under both the few-shot and full-shot settings. In particular, ZiRa surpasses CL-DETR and iDETR by substantial margins, with improvements of **13.91** and **8.74** AP under the full-shot setting in terms of "ZCOCO", respectively.

We then compare ZiRa with existing methods implemented on the OV-DINO model. The results in Tab. 2 show that ZiRa outperforms existing IOD methods in terms of "Avg" on downstream tasks. ZiRa also demonstrates excellent performance in preserving the zero-shot generalization ability of VLODMs under the full-shot setting. Specifically, ZiRa surpasses CL-DETR and iDETR by significant margins, with improvements of 14.55 and 11.36 AP in the full-shot setting for "ZCOCO," respectively.

These results emphasize ZiRa's effectiveness in maintaining the zero-shot AP and its potential as a superior solution for IOD. Distinctively, ZiRa, unlike CL-DETR and OW-DETR, requires minimal extra memory for a few branches without storing image exemplars and the model copy, making it more memory-efficient than other IOD methods.

### 4.3 Ablation Study

Table 3: Main ablation results.

| Rep+ | $L_{rdb}$ | $L_{hlrb}$ | ZCOCO | Avg | hAP | Ae | Aq | Co | Eg | Mu | Pa | Pv | Pi | Po | Ra | Sh | Th | Ve |
|---|---|---|---|---|---|---|---|---|---|---|---|---|---|---|---|---|---|---|
| ✗ | ✗ | ✗ | 39.72 | 58.12 | 47.19 | 30.26 | 47.11 | 69.30 | 66.16 | 56.82 | 58.20 | 64.34 | 69.08 | 48.48 | 63.35 | 41.53 | 76.59 | 64.37 |
| ✓ | ✗ | ✗ | 42.11 | 59.98 | 49.48 | 32.17 | 48.37 | 70.97 | 69.28 | 59.32 | 58.05 | **66.39** | **71.50** | 51.35 | 67.46 | 42.01 | 78.15 | 64.71 |
| ✗ | ✓ | ✓ | **46.09** | 54.24 | 49.83 | 30.14 | 33.84 | 68.80 | 65.29 | 42.27 | **60.20** | 61.42 | 69.54 | 34.79 | 64.99 | 39.31 | 74.41 | 60.12 |
| ✓ | ✓ | ✓ | 46.06 | 59.73 | **52.01** | **32.81** | 48.19 | 70.33 | 69.67 | 59.33 | 58.05 | 64.04 | 70.67 | 50.06 | 67.49 | 45.51 | 76.76 | 63.54 |
| ✓ | ✓ | ✗ | 46.01 | 58.43 | 51.48 | 30.77 | **48.91** | 69.69 | 68.93 | 51.98 | 58.05 | 61.80 | 70.58 | 45.71 | 67.72 | 43.35 | 78.19 | 63.95 |
| ✓ | ✗ | ✓ | 44.93 | **60.83** | 51.68 | 32.62 | 47.82 | **71.57** | **70.06** | **60.13** | 58.05 | 66.33 | 71.03 | **52.72** | **68.59** | **47.42** | **78.28** | **66.20** |

**Main Components of ZiRa.** Our approach combines the ZiL and the RDB to counteract forgetting in both the general domain and downstream tasks. In this study, we analyze the impact of these components. We utilize "Rep+" to denote whether to reparameterize the HLRB into LLRB after each task as well as using differentiated learning rates. Please note that even when we do not use "Rep+", we still retain the dual-branch structure like RDB. ZiL encompasses the RDB's ZiL ($L_{rdb}$) and HLRB's ZiL ($L_{hlrb}$), here we separate them and study their impact separately. We also evaluate performance with the "hAP" metric, which is the harmonic mean of "ZCOCO" and "Avg".

The results in Tab. 3 provide valuable insights. First, the comparison between the first and second rows demonstrates that "Rep+" can mitigate forgetting on both pre-training and downstream tasks. This effect comes from the labor division between branches that are built on differentiated learning rates and reparameterization. Second, the comparison between the first row and third rows highlights that ZiL ($L_{rdb} + L_{hlrb}$) significantly enhances the "ZCOCO". However, it cannot address forgetting on downstream tasks without "Rep+", and it even reduces the "Avg" since it also limits the plasticity

of the RDB. The best results lie in the fourth row, underscoring that combining "Rep+" and ZiL can effectively mitigate forgetting on both pre-training and downstream tasks to a greater extent. Conversely, the results in the last row illustrate that using ($L_{rdb}$) alone does not optimally address forgetting on downstream tasks. Comparing the second row and the last row, we can find that $L_{hlrb}$ can improve both "ZCOCO" and "Avg", showcasing that $L_{hlrb}$ can mitigate forgetting both on downstream tasks and pre-training, but without $L_{rdb}$, using $L_{hlrb}$ alone can not achieve the best "hAP".

**Learning on Different Modalities with ZiRa.** We carried out a series of experiments to investigate the effects of learning only on the vision side using parallel convolution layers (indicated as "V") and only on the language side using parallel linear layers (denoted as "L"). The results in Tab. 4 demonstrate that ZiRa can function independently on either the language or vision side, and it can also effectively function when simultaneously applied to both modalities, highlighting the generality of ZiRa's impact. Since the language and vision side learning acquires knowledge in distinct structures, neglecting either can hinder optimal learning. Consequently, simultaneously learning on both sides can outperform individual tuning on either the vision or language sides when employing ZiRa to avoid forgetting.

Table 4: Comparison of learning on different modalities.

| V | L | ZiRa | ZCOCO | Avg | hAP |
|---|---|---|---|---|---|
| ✗ | ✗ | ✗ | **47.37** | 46.70 | 47.03 |
| ✓ | ✗ | ✗ | 39.44 | 58.23 | 47.03 |
| ✗ | ✓ | ✗ | 45.83 | 54.77 | 49.90 |
| ✓ | ✓ | ✗ | 39.72 | 58.12 | 47.19 |
| ✓ | ✗ | ✓ | 45.96 | 57.64 | 51.14 |
| ✗ | ✓ | ✓ | 46.41 | 56.49 | 50.96 |
| ✓ | ✓ | ✓ | 46.06 | **59.73** | **52.01** |

Table 5: Comparison of different additional branch structures. We compared the results of introducing a single branch (denoted as SB), dual branches (denoted as DB), and RDB (DB + Rep+).

| Structure | ZCOCO | Avg | hAP |
|---|---|---|---|
| SB | **46.16** | 55.07 | 50.22 |
| DB | 46.09 | 54.24 | 49.83 |
| RDB | 46.06 | **59.73** | **52.01** |

**Branch Structure.** To study the impact of the additional branch structure introduced in VOLDM, we compared the results of introducing a single branch (denoted as SB), dual branches (denoted as DB), and RDB (DB + Rep+). Besides RDB, both the output norm of SB and DB are also penalized with ZiL. As we can see in Tab. 5, the SB structure, due to its inability to introduce inter-branch division of labor, falls short in its ability to prevent forgetting in downstream tasks compared to the RDB. Moreover, we can see that merely introducing DB with ZiL cannot prevent forgetting on downstream tasks, on the contrary, it brings worse forgetting due to increased plasticity. RDB not only incorporates the mechanism of branch labor division but also fully utilizes $l_{hlrb}$ to learn to minimize the interference of HLRB, thereby more effectively preventing the forgetting on downstream tasks.

# 5 Conclusion

This paper presents a novel learning task, Incremental Vision-Language Object Detection (IVLOD), which aims to continually adapt Vision-Language Object Detection Models (VLODMs) to multiple specialized domains while preserving VLODMs' zero-shot generalization ability. To solve this new task in a memory-efficient way, we introduce Zero-interference Reparameterizable Adaptation (ZiRa). ZiRa inserts Reparameterizable Dual Branche (RDB) on the both language side and vision side of the VLODM and constrains the RDB by Zero-interference Loss (ZiL) to protect the original generalizability of VLODMs and prevent forgetting on downstream tasks at the same time. Notably, ZiRa eliminates the need for saving the entire model copy for distillation or maintaining exemplars for replaying, which makes it a highly memory-efficient method. Extensive experiments conducted on the COCO and ODinW-13 datasets showcase the superiority of ZiRa for IVLOD.

**Limitations.** While ZiRa has demonstrated its effectiveness in IVLOD tasks, there are still some limitations. First, the current implementation of ZiRa is based on the DETR architecture, which may not be optimal for all VLODMs. Second, the current ZiRa method is designed for the IVLOD task, and its generalization to other incremental learning tasks remains to be explored.

## Acknowledgments

We would like to express our deepest gratitude to the anonymous reviewers and the community for their valuable feedback, which helped us enhance the quality of this work. Additionally, we are grateful to our colleagues for their insightful discussions and support throughout the development of this project.

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

# Appendix

**Effect of Hyperparameter**

**Impact of $\lambda$.** The hyperparameter $\lambda$ modulates the influence of ZiL within the overall loss function. In this study, we investigate the impact of varying values of $\lambda$ on model performance, as presented in Tab. 6. The results show that a small $\lambda$ value (e.g., $0.01$) tends to cause the model to overfit new tasks, resulting in reduced performance in the general domain. Conversely, a large $\lambda$ value (e.g., $1.0$) overly constrains the model, hindering its adaptation to downstream tasks. The optimal $\lambda$ value falls between these extremes, with the highest hAP observed when $\lambda = 0.05$, which strikes a balance between "ZCOCO" and "Avg".

Table 6: Results with varying $\lambda$ for ZiL.

| $\lambda$ | ZCOCO | Avg | hAP |
|------|-------|-------|-------|
| 0.00 | 39.72 | 58.12 | 47.19 |
| 0.01 | 44.58 | **60.86** | 51.46 |
| 0.05 | 45.65 | 60.54 | **52.05** |
| 0.10 | 46.06 | 59.73 | 52.01 |
| 0.20 | 46.25 | 58.13 | 51.51 |
| 0.50 | 46.60 | 54.27 | 50.14 |
| 1.00 | **46.94** | 50.08 | 48.46 |

**Differentiated Learning Rate.** We explore the effects of hyperparameter $\eta$. The results are detailed in Tab. 7. On the one hand, setting the LLRB's learning rate as zero equivalently freezes it. Although this results in the highest "Avg" by preserving downstream task knowledge, the "ZCOCO" performance suffers considerably. This degradation is because HLRB, which has a limited parameter size, endures all the optimization pressure from $L_{zil}$ and $L_{cls}+L_{loc}$. Despite HLRB's attempts to find a harmonious optimization path, it tends to emphasize downstream task learning over maintaining initial zero-shot generalizability, especially when the predominant loss $L_{cls} + L_{loc}$ outweighs $L_{zil}$. On the other hand, equating the LLRB's learning rate to the HLRB's is equivalent to creating two HLRBs inside the RDB. The enhanced plasticity that comes from the two-HLRB structure makes it challenging for ZiL to prevent it from gravitating toward directions detrimental to zero-shot generalizability, leading to diminished "ZCOCO" scores. Moreover, a significantly high LLRB learning rate can also precipitate a swift decline in downstream task knowledge retention. Optimal $\eta$ lies between these two cases, like $0.1$ or $0.2$, striking a balance between efficient new task adaptation and zero-shot generalizability protection.

Table 7: Results with varying $\eta$.

| $\eta$ | ZCOCO | Avg | hAP |
|------|-------|-------|-------|
| 0.00 | 44.97 | **60.23** | 51.49 |
| 0.10 | 45.93 | 59.79 | 51.95 |
| 0.20 | **46.06** | 59.73 | **52.01** |
| 0.50 | 45.87 | 59.46 | 51.79 |
| 0.70 | 45.65 | 58.78 | 51.39 |
| 1.00 | 44.82 | 58.64 | 50.81 |

**Impact of Different Initial Values of the Scaling Factor.** The scaling factor $s$ is a learnable parameter that affects the IVLOD's performance. Its initial value could shape the model's efficiency and adaptation capabilities. As documented in Tab. 8, contrary to initial expectations, a balanced scaling factor for both language and vision components, set at $1.00$, does not lead to the best outcomes. Instead, an asymmetrical approach where the vision component's scaling factor is reduced to $0.10$ while maintaining the language component's scaling factor at $1.00$, or vice versa, appears to enhance overall performance metrics, including "ZCOCO", "Avg", and "hAP". Remarkably, the configuration where both language and vision scaling factors are set to $0.10$ achieves superior results, yielding a slight improvement in "ZCOCO" and the highest "hAP" score among the tested scenarios. This suggests that a more conservative initial scaling factor may encourage the model to rely more on the base knowledge from pre-training, thus improving its generalization before adapting further through incremental learning.

Table 8: Impact of Different Initial Values of the Scaling Factor.

| Language | Vision | ZCOCO | Avg | hAP |
|---|---|---|---|---|
| 1.00 | 0.10 | 46.06 | 59.73 | 52.01 |
| 1.00 | 1.00 | 45.39 | 60.40 | 51.83 |
| 0.10 | 1.00 | 45.20 | 60.46 | 51.73 |
| 0.10 | 0.10 | 46.03 | 60.16 | 52.15 |

**Impact of the Norm Types of ZiL**

By default, we utilize the L1 norm to regulate ZiL. However, the choice of norm type for ZiL significantly impacts performance. As demonstrated in Tab. 9, L2 excels in ZCOCO, L1 strikes a balance between zero-shot and downstream performance, and Smooth L1 performs exceptionally well in downstream tasks while maintaining competitive zero-shot performance, even outperforming the L1 norm. Regardless of the norm type used, ZiRa remains effective, showcasing its versatility across different norm types.

Table 9: Results with Varying Norm Types of ZiL.

| Norm Type | ZCOCO | Avg | hAP |
|---|---|---|---|
| L1 | 46.06 | 59.73 | 52.01 |
| L2 | **46.26** | 57.32 | 51.20 |
| Smooth L1 | 46.10 | **60.07** | **52.17** |

**Visualization**

To more vividly showcase the effectiveness of the proposed ZiRa method, we also performed a visualization experiment. As shown in Fig. 6, using the pre-trained Grounding DINO directly to leverage its zero-shot detection capabilities allows for detecting many unseen objects, but fails to detect some. Upon incrementally fine-tuning with iDETR using training samples (with five classes per phase and one sample per class), the model will be able to recognize objects that are undetected by zero-shot detection. However, some objects that can be initially recognized by zero-shot detection are no longer detected. In contrast, after incremental fine-tuning with ZiRa, the images reveal that both familiar and novel objects are detected. This demonstrates ZiRa's dual ability to prevent forgetting both pre-trained knowledge and information from downstream tasks.

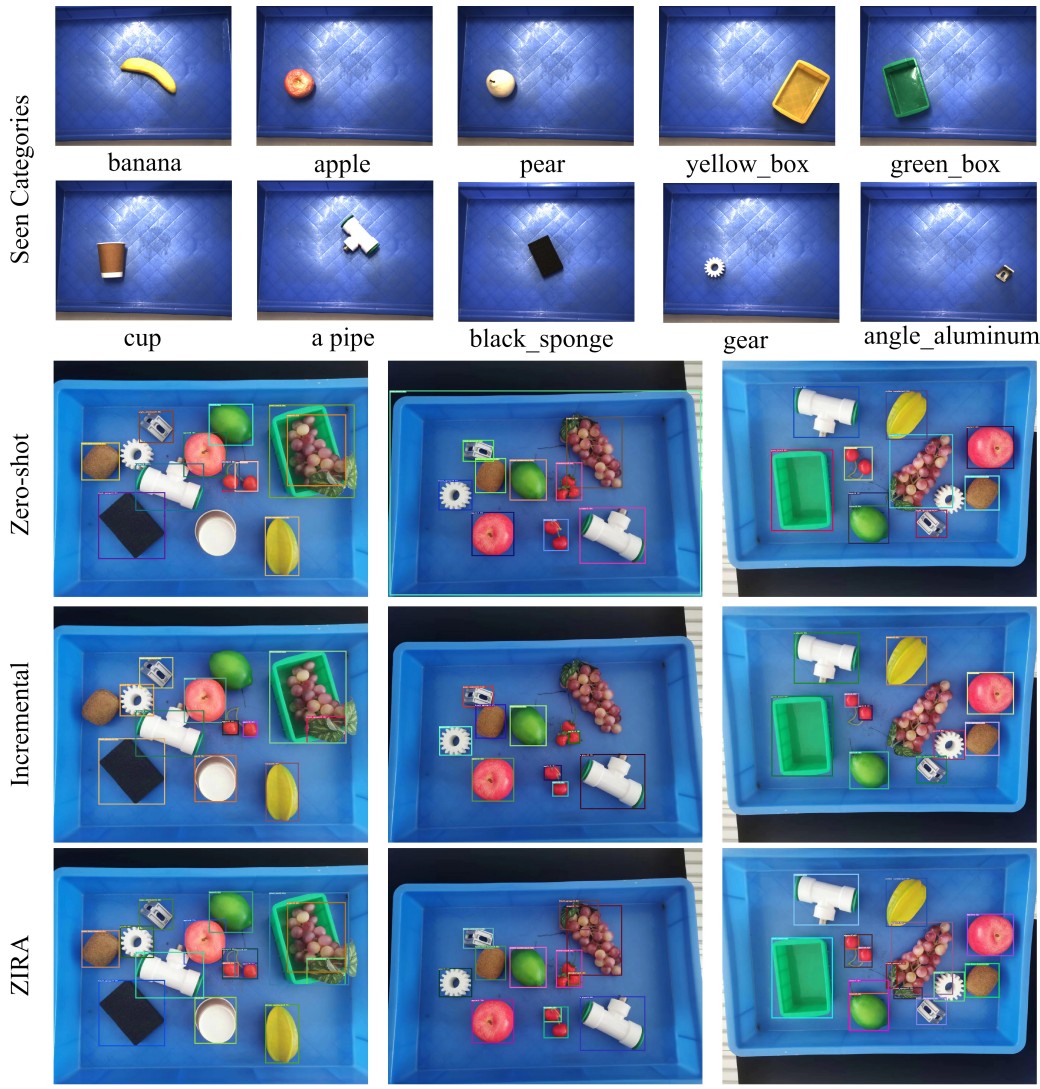

**Text Prompt : "a pipe.pear.banana.apple.green_box.yellow_box. grape.cup.gear.yellow carambola.lemon.cherry.black_sponge.brown kiwi fruit. angle_aluminum.strawberry"**

Figure 6: Visualization results of both seen categories and unseen categories with Grounding DINO-based ZiRa.

