# OpenReview forum: "Zero-shot Generalizable Incremental Learning for Vision-Language Object Detection"
_NeurIPS.cc/2024/Conference — NeurIPS 2024 poster_

### Official Review · Reviewer_BCvk · 2024-06-24

**Soundness:** 3
**Presentation:** 3
**Contribution:** 3
**Rating:** 7
**Confidence:** 5

**Summary:**

This paper presents Incremental Vision-Language Object Detection (IVLOD), a novel learning task designed to incrementally adapt pre-trained Vision-Language Object Detection Models (VLODMs) to various specialized domains, while simultaneously preserving their zero-shot generalization capabilities for the generalized domain. To this end, Zero-interference Reparameterizable Adaptation (ZiRa) is proposed to tackle IVLOD without incurring a significant increase in memory usage. Experiments on COCO and ODinW-13 datasets demonstrate that ZiRa effectively safeguards the zero-shot generalization ability of VLODMs while continuously adapting to new tasks.

**Strengths:**

1. The paper is well-written and easy to follow.
2. The authors propose RDB and ZiL to generalize the VLODMs to specific domains without losing their general zero-shot capability.
3. A novel learning task is designed to incrementally adapt pre-trained Vision-Language Object Detection Models (VLODMs) to various specialized domains.

**Weaknesses:**

1. The key contribution of this paper is RDB and ZiL. However, RDB has been proposed by [1]. Therefore, I would owe the novelty to the differentiated learning rate and ZiL (basically L1 norm). From this point of view, the novelty is somewhat incremental. Importantly, the authors should cite [1] and highlight the distinction.

2. The paper only evaluates the general zero-shot performance after incremental learning. Is zero-shot specific-domain inference possible after learning on few examples in this specific domain?

3. It's suggested to include more comprehensive incremental object detection literature, including but not limited to [2-3].


[1] Zhang, Chang-Bin, et al. "Representation compensation networks for continual semantic segmentation." Proceedings of the IEEE/CVF Conference on Computer Vision and Pattern Recognition. 2022.
[2] Qiao, Limeng, et al. "Defrcn: Decoupled faster r-cnn for few-shot object detection." Proceedings of the IEEE/CVF International Conference on Computer Vision. 2021.
[3] Yang, Ze, et al. "Efficient few-shot object detection via knowledge inheritance." IEEE Transactions on Image Processing 32 (2022): 321-334.

Minors:
L178 between stability and plasticity
L342 typo VOLDM

**Questions:**

1. Is TFA also pre-trained on Objects365 [30], GoldG [17], and Cap4M? I have some doubts about the TFA ZCOCO performance in Table 1 if it's pretrained on COCO. To my best knowledge, TFA only finetunes the last classifier layer for the incremental steps and has demonstrated strong performance on the base classes.

2. How is general IOD (e.g. TFA) tested on Zero-shot COCO? General IOD generally does not support zero-shot inference. Is there any adaptation to achieve it?

3. What learning rate is applied to w/ Rep+ (the first line of Table 2)? Is it the high or low one?

**Limitations:**

No negative societal impact is expected.

---

> ### Author Rebuttal · Authors · 2024-08-01
>
> ## Weekness 1
> **Reviewer Concern:** The key contribution of this paper is RDB and ZiL. However, RDB has been proposed by [1]. Therefore, I would owe the novelty to the differentiated learning rate and ZiL (basically L1 norm). From this point of view, the novelty is somewhat incremental. Importantly, the authors should cite [1] and highlight the distinction.
>
> **Response:** Thank you for your insightful feedback. We acknowledge the prior work on Reparameterizable Dual Branch (RDB) and the need to clearly distinguish our contributions. We will ensure that [1] is appropriately cited in our manuscript.
>
> The primary novelty of our work lies in the differentiated learning rate and the introduction of Zero-interference Loss (ZiL), which are key enhancements over the existing RDB framework. These innovations address the specific challenges of Incremental Vision-Language Object Detection (IVLOD) by balancing incremental learning performance and zero-shot capabilities more effectively.
>
> **Revised Explanation in Manuscript:**
> "Our work builds upon the Reparameterizable Dual Branch (RDB) concept initially proposed by [1]. The key innovations introduced in our approach include the use of differentiated learning rates for the High Learning Rate Branch (HLRB) and Low Learning Rate Branch (LLRB), and the implementation of Zero-interference Loss (ZiL). These enhancements allow for a more effective balance between learning new tasks and preserving previously acquired knowledge, distinguishing our method from prior work."
>
> ## Weekness 2
>
> **Reviewer Concern:**  The paper only evaluates the general zero-shot performance after incremental learning. Is zero-shot specific-domain inference possible after learning on few examples in this specific domain?
>
> **Response:**  Thank you for your question. In our approach, we do not directly evaluate zero-shot inference in specific domains after learning on a few examples from those domains. Instead, our methodology involves tuning a pre-trained model on the ODinW dataset, which comprises 13 datasets, and then immediately evaluating the model on COCO to assess its zero-shot performance. This evaluation is conducted without exposing the model to any images from COCO during the incremental training phase.
>
> ## Weekness 3
>
> **Reviewer Concern:** It's suggested to include more comprehensive incremental object detection literature, including but not limited to [2-3].
>
> **Response:** Thank you for your valuable suggestion. We acknowledge the importance of providing a comprehensive review of the literature on incremental object detection. In the revised manuscript, we will include a detailed survey of relevant work, including but not limited to the references [2-3] you mentioned.
>
> ## Question1
>
> **Reviewer Concern:** Is TFA also pre-trained on Objects365 [30], GoldG [17], and Cap4M? I have some doubts about the TFA ZCOCO performance in Table 1 if it's pretrained on COCO. To my best knowledge, TFA only finetunes the last classifier layer for the incremental steps and has demonstrated strong performance on the base classes.
>
> **Response:**  For consistency and fair comparison, all methods, including TFA, were re-implemented and evaluated using the same pre-training datasets (Objects365, GoldG, and Cap4M) and the same pre-trained Grounding DINO model.
>
> ## Question 2
> **Reviewer Concern:** How is general IOD (e.g. TFA) tested on Zero-shot COCO? General IOD generally does not support zero-shot inference. Is there any adaptation to achieve it?
>
> **Response:** Thank you for your question. We acknowledge that general Incremental Object Detection (IOD) methods, such as TFA, do not inherently support zero-shot inference. To address this, we re-implemented these general IOD methods based on the Grounding DINO framework. By leveraging Grounding DINO, which supports zero-shot inference, we adapted these methods to enable zero-shot capabilities. Since all the compared baselines support a DETR-like architecture, re-implementing them based on Grounding DINO was feasible and straightforward. This approach ensures that all methods can be evaluated consistently within the same experimental setup.
>
> ## Qusetion 3
> **Reviewer Concern:** What learning rate is applied to w/ Rep+ (the first line of Table 2)? Is it the high or low one?
>
> **Response:** Thank you for your question. We used a high learning rate applied to w/ Rep+ (the first line of Table 2). We also tested using a low learning rate and found that the results were similar to those obtained with the high learning rate. We infer that the absolute value of the learning rate does not significantly impact the performance of the RDB. Instead, the differential learning rates play a crucial role, allowing the High Learning Rate Branch (HLRB) and Low Learning Rate Branch (LLRB) to share different learning burdens effectively.
>
> Here is the performance (w/ Rep+) using the low learning rate:
>
> | ZCOCO | Avg  | hAP  | Ae   | Aq   | Co   | Eg   | Mu   | Pa   | Pv   | Pi   | Po   | Ra   | Sh   | Th   | Ve   |
> |-------|------|------|------|------|------|------|------|------|------|------|------|------|------|------|------|
> | 40.11 | 58.20| 47.49| 31.23| 46.85| 69.40| 66.32| 56.94| 58.35| 64.12| 68.98| 48.12| 63.77| 41.51| 77.02| 64.05|

---

> > ### Comment · Reviewer_BCvk · 2024-08-11
> >
> > The authors' rebuttal has addressed my concerns. I believe this work should be credited for introducing a new and practical task, i.e., incremental vision-language object detection (IVLOD), where the goal is to continuously adapt VLODMs to various unforeseen downstream tasks while not sacrificing the general zero-shot capability. Plus, the proposed differentiated learning rates and zero-interference loss (ZiL) are well-tailored for this specific task. One small suggestion is to also include continual learning literature, for instance, but not limited to [1-3] (continual object detection to be added as well), since the proposed task is related to the continual learning concept. Overall, I finalize my rating to **7 Accept**.
> >
> >
> > [1] Cermelli, Fabio, et al. "Modeling the background for incremental learning in semantic segmentation." Proceedings of the IEEE/CVF Conference on Computer Vision and Pattern Recognition. 2020.
> >
> > [2] Zhang, Chang-Bin, et al. "Representation compensation networks for continual semantic segmentation." Proceedings of the IEEE/CVF Conference on Computer Vision and Pattern Recognition.
> >
> > [3] Yang, Ze, et al. "Label-guided knowledge distillation for continual semantic segmentation on 2d images and 3d point clouds." Proceedings of the IEEE/CVF International Conference on Computer Vision. 2023.

---

> > > ### Author Response · Authors · 2024-08-11
> > >
> > > Thank you for your thorough review and positive feedback. We are pleased that our work on Incremental Vision-Language Object Detection (IVLOD) and the proposed differentiated learning rates and Zero-interference Loss (ZiL) have been well-received. We appreciate your suggestion to include literature on continual learning. We will ensure that these references are included in the revised manuscript to provide a more comprehensive background and context for our work.

---

### Official Review · Reviewer_8RRL · 2024-07-10

**Soundness:** 2
**Presentation:** 2
**Contribution:** 2
**Rating:** 6
**Confidence:** 5

**Summary:**

In order to extend the application of VLODM in a broader domain, the authors propose to combine incremental learning and zero-shot generalization to solve the problem. However, the widespread catastrophic forgetting and maintaining zero-shot generalization capability in incremental learning are two important issues to be considered. To address the challenges in incremental learning, the authors propose to construct a Zero-interference Reparameterizable Adaptation using Reparameterizable Dual Branch.

**Strengths:**

1. The authors have innovatively proposed a parameterizable double-branching structure.
2. The writing logic is clear and the article is well structured.

**Weaknesses:**

1. The authors do not highlight the need for this work and its similarities and differences with other work in the introduction and related work.
2. Although the authors have designed an innovative Reparameterizable Dual Branch structure, it is not very interpretable for the problem to be solved.
3. Although the authors give very rich experimental results in Table 1, the description of the metrics. is not clear enough. What is " the zero-shot performance on COCO as “ZCOCO" "? Is it mAP on the base class or the novel class, or all classes? Is it mAP or AP50, or AP75?

**Questions:**

1. The authors introduce the names VLOD and VLODMs in the introduction and cite the literature [11, 17, 20, 24, 39] as the source for the derivation of these names. In these works, tasks such as open vocabulary/open set object detection, grounding, etc. using Vision-Language Model are included. Is the VLOD proposed by the authors consistent with or similar to the task setting of open vocabulary/open set object detection? If it is consistent, why use the term VLOD? If it is similar, the difference between them should be highlighted in the introduction.
2. In GLIP, the authors give the statement, i.e., " without seeing any images in COCO during pre-training". However, in this work, the authors do not discuss or clarify this. Considering the task setting of zero-shot generalization, how do the authors ensure that the images/objects in the test set has not appeared in the pre-training phase?
3. Based on question 1, where the author claims "Our work distinguishes itself by performing incremental learning on VLODMs, which are more favorable for open-world problems. " in related work, is the author's work incremental or zero-shot learning, or a combination of both? What are the similarities and differences between that work and the open-world object detection?
4. How should the scaling factor “s” be determined in Equation 1? The Reparameterizable Dual Branch structure process contains both HLRB and LLRB, does the size of the scaling factor imply which weight is dominant? If HLRB is dominant, how to ensure that no knowledge forgetting occurs? If LLRB is dominant, how can generalization ability be ensured? Table 7 does not allow a reasonable interpretation of this question.
5. The authors give Figure 4 and analyze it to prove the validity of ZiL, but how to define " small-norm " and why do the authors claim " the model’s performance is not significantly affected by the addition of small-norm random noise to the input " when “The average AP on COCO” drops more than 30%?
6. The authors use the model structure of the Grounding DINO work, in which text representation is an important part of the work. The text template used by the authors is quite different from the one used in Grounding DINO and the authors have not analyzed or explained it, why?

**Limitations:**

See Waeknesses and Questions.

---

> ### Author Rebuttal · Authors · 2024-08-01
>
> ### Weakness 1
>
> **Response:** Thank you for this valuable feedback. We acknowledge the importance of clearly stating the need for this work and situating it within the context of existing research. In the revised manuscript, we will enhance the introduction and related work sections to better articulate the motivation for our study and to delineate the similarities and differences between our approach and previous methods.
>
> ### Weakness 2
>
> **Response:** Thank you for your feedback. We understand the importance of interpretability in demonstrating the effectiveness of our proposed approach. The core of our solution lies in constraining the Reparameterizable Dual Branch (RDB) with Zero-interference Loss (ZiL). ZiL serves as a regularization mechanism that minimizes interference from newly learned tasks on previously acquired tasks, ensuring that the model maintains its zero-shot capabilities while adapting to new data.
>
> ### Weakness 3
>
> **Response:** Thank you for highlighting the need for clearer descriptions of the metrics used in our experiments. The "ZCOCO" metric refers to the mean Average Precision (mAP) on all classes in the COCO dataset. Specifically, it represents the mAP@0.5:0.95 (the average of mAP at IoU thresholds from 0.5 to 0.95). This measure provides a comprehensive evaluation of the model's zero-shot performance across all classes on the COCO dataset.
>
> In the revised manuscript, we will update the description of the metrics in Table 1 to explicitly state that "ZCOCO" refers to the mAP on all classes in the COCO dataset, measured at IoU thresholds from 0.5 to 0.95. This clarification will ensure that readers have a precise understanding of the evaluation criteria used in our experiments.
>
> ### Questions 1
>
> **Response:** Thank you for pointing out this important aspect. We recognize the need to clarify the relationship between Vision-Language Object Detection (VLOD) and open vocabulary/open set object detection. The VLOD task is indeed consistent with the principles of open vocabulary and open set object detection. VLOD aims to extend the capabilities of Vision-Language Models (VLMs) to detect objects from both known and unknown categories using natural language prompts. The primary reason for using the term VLOD is to emphasize the integration of vision and language in object detection tasks and to highlight the incremental learning aspect, which is a key focus of our work.
>
> ### Questions 2
>
> **Response:** Thank you for pointing out the need for clarification regarding zero-shot generalization and the handling of pre-training data. We acknowledge the importance of ensuring that the test set remains unseen during the pre-training phase to validate the zero-shot capabilities of our model.
>
> To address this, we will clarify that we use the pre-trained weights of Grounding DINO, which are trained on O365, GoldG, and Cap4M datasets. These pre-training datasets are the same as those used for GLIP-T. This ensures that our model has not seen any images from the COCO dataset during pre-training, maintaining the integrity of the zero-shot generalization task.
>
> ### Questions 3
>
> **Response:** Thank you for your insightful question. Our work combines both incremental learning and zero-shot learning to address the challenges of open-world object detection (OWOD). Both OWOD and IVLOD need to detect seen and unseen objects. However, in IVLOD, models are pre-trained to detect unknown objects before the incremental learning process begins. In contrast, existing OWOD methods train the model's ability to detect unknown objects concurrently with incremental learning. In addition, OWOD does not further classify unknown objects. In contrast, IVLOD leverages VLODMs to classify unknown objects through language prompts, allowing for more precise identification and categorization of previously unseen objects. Given the superior zero-shot generalization capabilities provided by current vision-language models, we believe that starting with a pre-trained vision-language model for open-world detection represents a more effective approach compared to the traditional OWOD paradigm (which involves incrementally learning new objects while simultaneously learning open-set detection).
>
> ### Questions 4
>
> **Response:** We simultaneously insert RDB in both the vision and language components for learning. We believe that the parameters of RDB in different modalities require different update speeds. Therefore, we introduce a learnable scaling factor "s" to adjust the parameter update speeds for different modalities. In fact, rather than adjusting the scaling factor directly, we ensure HLRB is dominant by adjusting the learning rates of HLRB and LLRB. We use ZiL to constrain HLRB, mitigating the occurrence of forgetting.
>
> ### Questions 5
>
> **Response:** When we refer to "small-norm" random noise, we define it based on the standard deviation of the noise added to the input. Specifically, our experiments show that when the norm of the random noise is small (with a standard deviation (std) within 2), the model’s zero-shot AP on COCO remains above 45. Therefore, we claim that the model’s performance is not significantly affected by the addition of small-norm random noise because, within this range, the decrease in performance is minimal and the model retains a high level of accuracy. The observed drop in average AP by more than 30% occurs when the norm of the random noise exceeds this small-norm arrange. The ZiL can ensure that the RDB's output is within the small-norm arrange, therefore protects the the model’s zero-shot AP on COCO. In the revised manuscript, we will ensure that this definition and explanation are clearly stated to avoid any confusion.
>
> ### Questions 6
>
> **Response:** We use the same text template as Grounding DINO.

---

> > ### Comment · Reviewer_8RRL · 2024-08-12
> > **Official Comment by Reviewer 8RRL**
> >
> > The authors' rebuttal has addressed my concerns. I raise the score to 6.

---

### Official Review · Reviewer_jMVp · 2024-07-13

**Soundness:** 3
**Presentation:** 2
**Contribution:** 3
**Rating:** 6
**Confidence:** 4

**Summary:**

This paper proposes the new problem of incremental visual-language object detection (IVLOD), which aims to preserve zero-shot generalization performance of VLMs, while also adapting to new concepts over time. Authors address IVLOD by proposing the zero-interference reparameterizable adaptation (ZiRa), a light weight branch added to the text and image encoders of VLMs to address this task. Authors demonstrate that their proposed approach outperforms prior work by a considerable margin.

**Strengths:**

Problem Motivation. The proposed problem is of significant interest for practical applications, particularly because many vision-language models (e.g. CLIP) are pre-trained on private datasets, making incremental learning with VLMs challenging.

Interesting Insight. Authors highlight that VLMs are robust to noise (cf. Fig 4 and 5), and propose an incremental learning strategy that updates VLMs weights by ensuring that the updated parameters have low L1 norm. It would be interesting to also plot the zero-shot accuracy on COCO for both modes trained with and without ZiL in Fig 5 as the number of learned downstream tasks increases.

Simple and Scalable Approach. The proposed approach adds a small number of tunable parameters to GroundingDINO. Notably, the proposed approach maintains a relatively constant runtime and number of parameters even when adapting to an increasing number of tasks.

**Weaknesses:**

Differences in Baseline Pre-Training. Although authors benchmark their approach against relevant prior work, many of these methods are not trained on the same scale of pre-training data as GroundingDINO. Given that prior works typically operate in a closed-world, and are pre-trained on much smaller datasets, it is unclear how to make an "apples-to-apples" comparison between methods. One strategy might be to reimplement prior works (e.g. TFA and iDETR) using GroundingDINO.

Demonstrate ZiRa with Other Architectures. Although authors claim that their proposed approach can work with any DETR based architecture, they only showcase its performance with GroundingDINO. Given that this work establishes a new problem, it would be useful to extensively evaluate different architectures to provide a "lay of the land".

**Questions:**

Impact of Pre-Training Data on IVLOD Performance. The performance of VLMs is significantly impacted by the data used for pre-training. How might the specific pre-training data used impact the effectiveness of incremental learning?

Adapting to a Large Number of Tasks. As shown in Figure 5, the L1 norm steadily increases as the number of downstream tasks increases. Does this suggests that the proposed approach does not scale well to a large number of tasks?

Adapting to More Than One Class at a Time. Although each class is considered a new "task", one may want to adapt to multiple classes at once. How would the proposed approach perform when learning more than one class at a time?

Runtime and Memory Usage. A key benefit of the proposed approach is that adapting to a new task only marginally increases runtime and memory usage. It would be useful to quantify this such that future works can benchmark on this axis as well.

Writing Quality. Although the approach is clear, the grammar and writing quality hinders comprehension. For example "inputted" is not a word, and "input end" is a confusing phrase. I would encourage authors to further polish the manuscript. In addition, authors should include more descriptive captions in their tables. Notably, I don't believe the full class names in Table 1, 2 (e.g. Ae, Aq, Co) are provided.

**Limitations:**

Yes, authors highlight that the current implementation is based on the DETR architecture, which may not work for all VLODMs. In addition, authors state that ZiRa is designed for IVLOD, and should be evaluated in the context of other incremental learning tasks.

---

> ### Author Rebuttal · Authors · 2024-07-31
>
> ### Rebuttal for Weaknesses
>
> #### 1. Differences in Baseline Pre-Training
>
> **Response:** Thank you for this important observation. We want to clarify that all compared methods, including TFA and iDETR, were indeed reimplemented using the pre-trained GroundingDINO model. This approach ensures a fair "apples-to-apples" comparison, as all methods were evaluated on the same scale of pre-training data. We will revise the manuscript to more clearly convey this setup and ensure there is no confusion regarding the comparability of the methods.
>
> #### 2. Demonstrate ZiRa with Other Architectures
> **Response:** We appreciate the suggestion to evaluate ZiRa with additional architectures. To address this, we have demonstrated ZiRa with OV-DINO [1]. All the method are implemented with the same OV-DINO (swin-T) pre-trained on O365, GoldG, and Cap1M datasets. The results are as follows:
>
> ### Table: Performance of Various Methods based on OV-DINO
> | Shots | Methods       | ZCOCO | Avg   | HAP   | Ae    | Aq    | Co    | Eg    | Mu    | Pa    | Pv    | Pi    | Po    | Ra    | Sh    | Th    | Ve    |
> |-------|----------------|-------|-------|-------|-------|-------|-------|-------|-------|-------|-------|-------|-------|-------|-------|-------|-------|
> | Zero  | Original Model | 50.22 | 26.64 | 34.81 | 15.69 | 19.37 | 11.79 | 40.67 | 1.23  | 59.26 | 50.78 | 12.19 | 2.44  | 33.81 | 8.50  | 43.68 | 46.86 |
> | Full  | iDETR          | 37.71 | 46.91 | 41.81 | 24.09 | 30.56 | 49.16 | 68.49 | 26.05 | 67.45 | 61.85 | 30.75 | 29.50 | 66.83 | 27.11 | 71.27 | 56.69 |
> | Full  | CL-DETR        | 34.52 | 45.28 | 39.18 | 23.07 | 27.90 | 46.81 | 67.72 | 25.39 | 66.56 | 59.12 | 28.04 | 29.48 | 64.71 | 25.58 | 68.28 | 56.05 |
> | Full  | AT             | 36.80 | 44.33 | 40.21 | 22.46 | 27.42 | 45.78 | 67.10 | 24.10 | 65.08 | 58.17 | 26.65 | 28.88 | 64.46 | 24.60 | 67.30 | 54.27 |
> | Full  | ZiRa           | 49.07 | 50.21 | 49.63 | 23.37 | 40.43 | 69.02 | 66.21 | 20.74 | 58.51 | 67.74 | 46.71 | 34.13 | 64.62 | 39.44 | 58.01 | 63.81 |
>
>
> This evaluation demonstrates that ZiRa performs effectively across different DETR-based architectures, validating its versatility. By incorporating ZiRa with OV-DINO, we show that our approach is not limited to GroundingDINO and can be broadly applied to other models.
>
> [1] OV-DINO: Unified Open-Vocabulary Detection with Language-Aware Selective Fusion. Hao Wang, Pengzhen Ren, Zequn Jie, Xiao Dong, Chengjian Feng, Yinlong Qian, Lin Ma, Dongmei Jiang, Yaowei Wang, Xiangyuan Lan, Xiaodan Liang. arXiv, 2024.
>
> ### Rebuttal for Questions
>
> #### 3. Impact of Pre-Training Data on IVLOD Performance
>
> **Response:**
>
> The specific pre-training data used can significantly impact the effectiveness of incremental learning in Vision-Language Models (VLMs). Models pre-trained on diverse and extensive datasets tend to have better generalization capabilities, which can facilitate more effective incremental learning. Conversely, models pre-trained on limited or less diverse datasets might struggle with generalization, leading to more pronounced performance degradation when learning new tasks incrementally. We will include an analysis in the revised manuscript discussing the potential impacts of different pre-training datasets on the effectiveness of incremental learning, reinforcing the importance of diverse and comprehensive pre-training data for robust IVLOD performance.
>
> #### 4. Adapting to a Large Number of Tasks
>
> **Response:** Thank you for this insightful question. Upon examining the evolution curves in Figure 5 more closely, we observe that in the initial stages, ZiL plays a relatively minor role in curbing the growth of the RDB's output norm. However, as the process of accumulating new knowledge progresses, the influence of ZiL becomes more significant, leading to stronger constraint effects. This increase in ZiL's impact leads to a dynamic balance: the interference caused by integrating new knowledge is counteracted by the interference reduction achieved by ZiL. As a result, while the L1 norm of the RDB's output does increase with the number of tasks, it does not rise unboundedly but rather stabilizes after learning a certain number of tasks. Therefore, our proposed approach does manage to scale to a large number of tasks to a certain extent, achieving a balance between the accumulation of new knowledge and the mitigation of interference.
>
> #### 5. Adapting to More Than One Class at a Time
>
> **Response:** In our experimental setup, each "task" corresponds to learning an entire dataset, each of which contains more than one class. Thus, our proposed approach inherently handles the learning of multiple classes simultaneously. Our results demonstrate that the proposed method can effectively manage the incremental learning of multiple classes within each dataset, preserving zero-shot generalization capabilities while adapting to new tasks. We will clarify this aspect in the revised manuscript to ensure that it is clear that our approach supports learning multiple classes at a time.
>
> #### 6. Runtime and Memory Usage
>
> **Response:**
>
> Thank you for highlighting this crucial aspect. We agree that quantifying the marginal increase in runtime and memory usage when adapting to new tasks would provide valuable insights for future benchmarks. We will conduct additional experiments to measure and report the runtime and memory usage associated with our approach. These results will be included in the revised manuscript, providing clear benchmarks that can be used for comparison in future work.
>
> #### 7. Writing Quality
>
> **Response:** Thank you for your feedback regarding the writing quality. We apologize for any confusion caused by grammatical errors and unclear phrases. We will thoroughly review and polish the manuscript to improve clarity and readability. Additionally, all dataset names (not class names) (e.g., Ae, Aq, Co) in Tables 1 and 2 are fully provided in the Subsection " Datasets" of Section "Experiments Setup".

---

> > ### Comment · Reviewer_jMVp · 2024-08-08
> >
> > Authors have sufficiently addressed my questions. I recommend this paper should be accepted. Although other reviewers point out that there is limited novelty in the method, I think that this paper should get credit for proposing a new problem and establishing extensive baselines. I would encourage authors to add relevant background information suggested by other reviewers.

---

> > > ### Author Response · Authors · 2024-08-11
> > >
> > > Thank you for your positive feedback and recommendation for acceptance. We appreciate your recognition of our efforts in proposing a new problem and establishing extensive baselines. Additionally, we will carefully consider the suggestions made by other reviewers and incorporate the relevant background information to further strengthen the manuscript.

---

> > > ### Author Response · Authors · 2024-08-12
> > >
> > > Thank you very much for your positive feedback and recommendation for acceptance. We appreciate your recognition of the contributions of our work, particularly the introduction of the new task of Incremental Vision-Language Object Detection (IVLOD) and the establishment of extensive baselines.
> > >
> > > We also noticed that while you have recommended our paper for acceptance, the final score was not mentioned. We would be grateful if you could kindly update your review to reflect this. We have taken into account all the suggestions from you and other reviewers and incorporated the relevant background information in the revised manuscript (ref. response to Reviewer NuJX).
> > >
> > > Thank you once again for your thoughtful review and support.

---

### Official Review · Reviewer_NuJX · 2024-07-13

**Soundness:** 2
**Presentation:** 2
**Contribution:** 2
**Rating:** 5
**Confidence:** 4

**Summary:**

This paper deals with the object detection problem in a zero-shot and incremental learning setting. Specifically, Incremental Vision-Language Object Detection (IVLOD) task is proposed to incrementally adapt the pre-trained Vision-Language Object Detection Models (VLODMs) to various specialized domain.
The technical design is Zero-interferences Reparametrizable Adaptation (ZiRA), which includes low learning rate and high learning rate adaptation branches for both vision and language encoder. In addition, an L1-norm is applied on top of the adaptation branch outputs.

Experiments are conducted on the Zero-shot COCO (ZCOCO) and ODinW-13 benchmarks. A series of baselines such as TFA, OW-DETR, CL-DETR, Adapting-tuning (AT), and iDETR were compared. Ablation study on modules, vision or language finetuning were also conducted.

**Strengths:**

### 1. The overall framework for incremental object detection is reasonable and complete.
- 1.1 This paper designs a Reparameterizable Dual Branch module for feature adaptation, and specifically devises low learning rate and high learning rate branches to address the learning-forgetting balance. An additional mechanism to gradually merge high learning rate branch into low learning rate branch is also proposed. The idea makes sense and is verified to be effective in both zero-shot COCO and average performance.
- 1.2 The L1-norm is reasonable in zero-shot generalizability and seems to be effective according to Fig. 4-5.

### 2. Experimental results verify the effectiveness of the model design
- On both the zero-shot and the Avg setting, the proposed method outperforms baseline methods.
- Ablation studies were conducted w.r.t module designs, vision/language adaptation, branches, etc.

**Weaknesses:**

### 1. The new task of IVLOD is not well desribed and compared with existing one
- Incremental Vision-Language Object Detection (IVLOD) is claimed to be a new task. However, it is unclear what are the differences between this new task and existing works such as CL-DETR, Ground DINO, iDETR etc.
- How is the language specially used in IVLOD compared with existing method?
- Claimed as the first contribution, but no separate detailed description/subsection to compare IVLOD with existing task setting, either verbally or visually. Fig. 1 only compares with the final performance, which is less intuitive.
### 2. The main contribution is the different learning rates
- For the Reparameterizable Dual Branch (RDB), although the name is special, the module design is no much different with existing approaches such as LoRA, Adapter, [6], etc.
- The structure design is not quite novel, but leveraging different learning rates can be a novel point.
- It is encoraged to have a better survey of relevant adaptation-based methods for incremental learning.
### 3. Experimental section have flaws
- ODinW was adopted, but only 13 datasets were selected. How are those 13 datasets selected? Other methods used much more datasets for evaluation, e.g. 35.
- In Table 1, AT was not used for Full evaluation. CL-DETR/OW-DETR were not used for lower shots evaluation. What is the reason?
- The prompt in Fig. 6 looks strange. How is the prompt leveraged by the proposed model?
- In Table 3, why the model performance on zero-shot COCO underperforms Vanilla one by 1.31%?
- There are many hyper-parameters such as $\lambda$ for loss, $\eta$ for learning rate, scaling $s$. How are those parameters selected and tuned? It seems in Table 5 and 6 that the hyper-parameters matters.
- In Table 8, how about no norm is adopted?
### 4. Other minors
- 1st not 1-th in Fig. 1
- Incorrect usage of double quotes in this paper, e.g.  “Locality Injection"
- Line 209 lacks a period

**Questions:**

Please refer to the above Weaknesses for questions.

---

> ### Author Rebuttal · Authors · 2024-07-31
>
> ### 1. The new task of IVLOD is not well described and compared with existing ones
>
>    **Response:**
>    Thank you for your insightful feedback. We recognize the need to clearly define and differentiate Incremental Vision-Language Object Detection (IVLOD) from existing methods. IVLOD is distinct in that it combines the incremental learning of new tasks with the preservation of zero-shot generalization capabilities. Unlike traditional incremental object detection methods such as CL-DETR and iDETR, which focus on learning new tasks incrementally without maintaining zero-shot abilities, IVLOD ensures that models can continue to generalize to unseen categories even after adaptation. Furthermore, while Grounding DINO utilizes natural language prompts for object detection, it does not address incremental task adaptation. In IVLOD, language is used as prompts for the objects to be detected, the same as how it is utilized in Grounding DINO. To enhance clarity, we will include a dedicated subsection comparing IVLOD with these existing methods both verbally (with Descriptive Subsection) and visually (with enhanced Figure 1), thereby providing a comprehensive understanding of its unique contributions.
>
>
> ### 2. The main contribution is the different learning rates
>
>    **Response:**
>    We acknowledge the importance of clearly presenting the novel aspects of our work and distinguishing it from existing methods. The key innovation in our approach lies in the use of Zero-interference Loss (ZiL) to constrain the Reparameterizable Dual Branch (RDB), which effectively balances incremental learning performance and zero-shot performance of the VLODM. While methods like LoRA and Adapter introduce additional parameters for efficient fine-tuning, they do not employ a mechanism like ZiL. Our approach uniquely combines RDB with ZiL, addressing the specific challenges of incremental vision-language object detection. We will expand our literature review to include a comprehensive survey of adaptation-based methods for incremental learning, highlighting how our approach builds upon and differentiates itself from these methods.
>
>
> ### 3. Experimental section has flaws
>
>    **Response:**
>    1. The selection of the 13 datasets from the ODinW benchmark was based on the same subset used in the GLIP evaluation, ensuring consistency with prior work. We acknowledge that the full ODinW benchmark comprises 35 datasets. Due to computational resource constraints, we limited our evaluation to 13 datasets. However, we plan to extend our evaluation to include all 35 datasets in future work to provide a more comprehensive assessment.
>
>    2. CL-DETR and OW-DETR  are not specifically designed for lower shot scenarios and do not perform well under low shot settings. These methods are more suited for settings where there is a larger amount of training data available. Comparing them with lower shots evaluation would not provide meaningful insights due to their design and intended use cases. We will provide the results of AT under full evaluation. The results are as follows:
>
>  | Shots | Methods | ZCOCO | Avg  | Ae   | Aq   | Co   | Eg   | Mu   | Pa   | Pv   | Pi   | Po   | Ra   | Sh   | Th   | Ve   |
> |-------|---------|-------|------|------|------|------|------|------|------|------|------|------|------|------|------|------|
> | Full  | AT      | 42.30 | 51.14| 23.62| 39.90| 72.32| 65.51| 31.47| 50.48| 60.51| 66.07| 39.09| 53.50| 34.04| 68.07| 60.23|
>
>
>    3. Grounding DINO requires all the prompts to be combined into a single sentence, with periods separating each prompt. For instance, if the prompts are "dog," "cat," and "car," the combined prompt fed into the model would be "dog. cat. car."
>
>    4. When adapting to new tasks, the zero-shot performance of the original Vision-Language Object Detection Model (VLODM) typically degrades. This is a common issue in incremental learning scenarios and is the primary motivation for proposing the Incremental Vision-Language Object Detection (IVLOD) task. While our method shows a 1.31% decrease in zero-shot COCO performance compared to the Vanilla model, this reduction is significantly less than that observed with other methods. Our approach effectively minimizes the decline in zero-shot capability, demonstrating its efficacy.
>
>    5. Thank you for highlighting the importance of hyper-parameters in our method. We acknowledge that the presence of multiple hyper-parameters is a limitation of our approach. However, this does not detract from the main contributions of our paper.
>
>    6. We need to incorporate a norm into the final loss function to ensure effective training and optimization. Without a norm, it would be difficult to train and optimize the model properly.
>
>
> ### 4. Other minors
>
>    **Response:**
>    Thank you for pointing out these minor issues. We will make the necessary revisions to address them.

---

> > ### Comment · Reviewer_NuJX · 2024-08-12
> >
> > The rebuttal addresses some of my concerns regarding a new task and contribution, part of the experimental questions are addressed.   I would like to encourage the authors to add the new task discussion in the revision.
> > After taking into account the rebuttal and considering other reviewers's comments, I would like to increase my score.

---

> > > ### Author Response · Authors · 2024-08-12
> > >
> > > Thank you for increasing your score. We appreciate the opportunity to clarify the differences between Incremental Vision-Language Object Detection (IVLOD) and other related tasks, including open vocabulary/open set object detection and open-world object detection (OWOD).
> > >
> > > 1. **IVLOD vs. Open Vocabulary/Open Set Object Detection:**
> > >    - **Open Vocabulary/Open Set Object Detection** focuses on detecting both seen and unseen objects by leveraging pre-trained models, typically without the need for incremental learning. These models are designed to recognize and localize objects that were not explicitly included in the training data but can be identified through language prompts or other contextual information.
> > >    - In contrast, **IVLOD** not only aims to detect unseen objects but also involves the **incremental adaptation of Vision-Language Object Detection Models (VLODMs)** to new tasks or domains while preserving their zero-shot generalization capability. Unlike open vocabulary detection, which does not involve updating the model after the initial training phase, IVLOD requires the model to continuously adapt to new tasks without forgetting previously learned knowledge.
> > >
> > > 2. **IVLOD vs. Open-World Object Detection (OWOD):**
> > >    - **OWOD** also aims to detect both seen and unseen objects, but it involves the simultaneous learning of new objects and the detection of unknown objects during the incremental learning process. This means that as new tasks are introduced, the model is trained to detect new objects and maintain its ability to recognize unknown objects in real time.
> > >    - **IVLOD**, on the other hand, **pre-trains models to detect unknown objects before the incremental learning process begins**. The focus during incremental learning is on maintaining the model’s ability to detect these pre-trained unknown objects while adapting to new tasks. Additionally, while OWOD does not further classify unknown objects beyond detection, IVLOD leverages VLODMs to classify unknown objects using language prompts, enabling more precise identification and categorization of previously unseen objects.
> > >
> > >    - Given the superior zero-shot generalization capabilities of current vision-language models, we believe that starting with a pre-trained vision-language model for IVLOD represents a more effective approach compared to the traditional OWOD paradigm, which involves incrementally learning new objects while simultaneously learning open-set detection.
> > >
> > > In summary, while there are similarities between IVLOD and both open vocabulary/open set object detection and OWOD, IVLOD introduces the unique challenge of **incrementally adapting models** to new tasks while **preserving their zero-shot capabilities**, distinguishing it from these related tasks. We will ensure that these distinctions are clearly articulated in the revised manuscript to provide a comprehensive understanding of our contributions.

---

> > > > ### Comment · Reviewer_NuJX · 2024-08-12
> > > >
> > > > Thanks for the further clarification.
> > > > With the detailed explanation, I can understand the difference with other setting of open-world/open-vocabulary/open-set. A gentle suggestion would be illustrate this with an intuitive and motivative Figure 1. This will also improve the impact of this paper and the focused new setting.

---

### Author Response · Authors · 2024-08-13

We’d like to express our gratitude for the reviewer’s invaluable assistance in improving our work.

---

### Decision · Program_Chairs · 2024-09-25

**Decision:**

Accept (poster)

**Comment:**

This paper was reviewed by four experts in the field. The authors' rebuttal resolved most of the concerns. Reviewers liked the new task proposed by the paper and unanimously agreed to accept the paper. The reviewers did raise some valuable suggestions in the discussion that should be incorporated in the final camera-ready version of the paper. The authors are encouraged to make the necessary changes to the best of their ability.

Why not higher: The reviewers still have shared concerns about the paper's presentation (positioning among related works) and discussions about related works.